# Introducing BPaL: Experiences from countries supported under the LIFT-TB project

D. F. Wares[1]*, M. Mbenga[1], V. Mirtskhulava[1], M. Quelapio[1], A. Slyzkyi[1‡], I. Koppelaar[2‡], S. N. Cho[3], U. Go[3], J. S. Lee[3], J.-K. Jung[3], D. Everitt[4], S. Foraida[4], M. Diachenko[4], S. Juneja[4], E. Burhan[5‡], A. Totkogonova[6‡], Z. Myint[7‡], I. Flores[8‡], N. A. Lytvynenko[9‡], N. Parpieva[10‡], N. V. Nhung[11‡], A. Gebhard[1]

**1** KNCV Tuberculosis Foundation, The Hague, The Netherlands, **2** Lelie Care Group, Rotterdam, The Netherlands, **3** International TB Research Center, Changwon, Republic of Korea, **4** TB Alliance, New York City, NY, United States of America, **5** Persahabatan Hospital, Jakarta, Indonesia, **6** National Center of Phthisiology, Bishkek, Kyrgyzstan, **7** National TB Programme, Yangon, Myanmar, **8** Jose B Lingad Memorial Regional Hospital, San Fernando City, The Philippines, **9** National Institute of Phthisiology and Pulmonology, Kyiv, Ukraine, **10** Republican Specialized Scientific and Practical Medical Center of Phthisiology and Pulmonology, Tashkent, Uzbekistan, **11** National Lung Hospital, Hanoi, Viet Nam

☯ These authors contributed equally to this work.
‡ AS, IK, EB, AT, ZM, IF, NAL, NP and NVN also contributed equally to this work.
* fraser.wares@kncvtbc.org

**Data Availability Statement:** The data cannot be shared publicly because of legal ownership restrictions. All data is the property of the respective countries. The authors have access to

## Abstract

### Background

Previously, drug-resistant tuberculosis (DR-TB) patients were treated with long, toxic, and relatively ineffective regimens. However, in recent years, there have been major improvements made. The 2020 World Health Organization DR-TB Treatment guidelines recommended the use of a 6-months all-oral BPaL (bedaquiline, pretomanid and linezolid) regimen under operational research (OR) conditions for selected DR-TB patients.

### Methods

The processes, challenges, and interim results of introducing BPaL under OR conditions in 7 countries supported under the Korea International Cooperation Agency/TB Alliance-funded "Leveraging Innovation for Faster Treatment of Tuberculosis (LIFT-TB)" project are described here. The OR objectives were to explore the feasibility of introducing the BPaL regimen, and to estimate its effectiveness and safety in a select group of DR-TB patients.

### Results

Between November 2020 and the end of March 2023, a total of 574 patients had been enrolled. Interim treatment success stands at an encouraging 90.9% (280/308). Although adverse events of special interest (AESI) were common, they were manageable, and only 1 patient had to discontinue the complete BPaL treatment regimen. In addition, no unexpected adverse events (AE) were seen.

the respective databases via data sharing agreements signed with the Ministry of Health of each country. The data underlying the results presented in the study, can be made available if permission is granted by the respective Ministry of Health for researchers who meet the criteria for access to confidential data. Kyrgyzstan: National TB Programme, 90 A Ahunbaeva str, Bishkek 720075, Kyrgyz Republic. Indonesia: National TB Programme, Ministry of Health, Republic of Indonesia, JI. HR Rasuna Said Blok X5 Kav.4-9, Setiabudi, Jakarta Selatan, Jakarta 12950, Indonesia. Myanmar: National TB Programme, Department of Public Health, Ministry of Health and Sports, Naypyidaw, Myanmar. The Philippines: Department of Health, San Lazaro Compound, Tayuman, Sta. Cruz, Manila, The Philippines. Ukraine: Public Health Center of the Ministry of Health of Ukraine, Kyiv, Ukraine. Uzbekistan: National TB programme, Republican Specialized Scientific-Practical Medical Center of Phthisiology and Pulmonology under Ministry of Health of the Republic of Uzbekistan, 1 Majlisiy str., Tashkent 100086, Uzbekistan. Vietnam: National TB Programme, National Lung Hospital, 463 Hoang Hoa Tham, Ba Dinh, Hanoi, Vietnam.

**Funding:** Activities under the "Leveraging Innovation for Faster Treatment of Tuberculosis (LIFT-TB)" project in the seven countries and the assistance provided from the global levels of KNCV TB Foundation, the Netherlands and the International TB Research Center, Republic of Korea, global levels, were supported by funding from the Korea International Cooperation Agency (KOICA), Republic of Korea and the TB Alliance (TBA), the United States of America. There was no additional external funding received for this study. The following authors received partial salary support via the LIFT-TB project: DFW; MM; VM; MQ; AS; IK; SNC; UG; JSL; J-KJ; and AG. KOICA had no role in the operational research study design, data collection and analysis, decision to publish, or preparation of the manuscript. As lead co-ordinating partner in the LIFT-TB project, TBA did play a role in the operational research study design, data collection and analysis, decision to publish, and preparation of the manuscript.

**Competing interests:** I have read the journal's policy and the following co-authors of this manuscript -S. Foraida, M. Diachenko, and S. Juneja have the following competing interests: Paid employment or consultancy with TB Alliance, the developer of pretomanid. This does not alter our adherence to PLOS ONE policies on sharing data and materials.

## Conclusion

With careful advocacy, frequent communication with partners, and following steps to strengthen essential aspects of the delivery system, the project's experiences show that BPaL OR was feasible across different country settings. Project documents were constantly updated. The sharing of information, experiences, and interim results had a significant positive and motivating effect within and across countries. Interim OR results show excellent patient responses and are comparable to those seen under trial conditions. Although common, the observed AEs and AESIs were manageable, and no unexpected AEs were seen.

## 1. Introduction

Drug-resistant tuberculosis (DR-TB) continues to be a public health threat. In 2020, it was estimated that only about 1 in 3 of the people who developed multidrug-resistant TB (MDR-TB–TB strain resistant to at least isoniazid and rifampicin) or rifampicin-resistant TB (RR-TB), were enrolled onto treatment [1, 2]. Due to the impact of the COVID-19 pandemic, numbers in 2020 and 2021 in fact fell compared with previous years [1, 2]. For many years, MDR-/RR-TB patients had been treated with a World Health Organization (WHO) recommended conventional MDR-TB regimen, which generally had an 8-month intensive phase of treatment, including a second-line injectable (SLI) agent, and a total duration of treatment of 20 months. However, in recent years, there have been major improvements in the treatment of MDR-/RR-TB. Since 2016, the WHO has recommended that for certain MDR-/RR-TB patients, a shorter 9–12 months treatment regimen (STR), with a SLI agent included, could be used instead of a longer (preferably all-oral) regimen [3]. From 2018 onwards, WHO has considered that a 9-month STR with bedaquiline (Bdq) replacing the SLI agent could be used [4].

A 6 months all-oral regimen, containing a new drug–pretomanid (Pa), in combination with bedaquiline (Bdq) and linezolid (Lzd), was trialed in the TB Alliance (TBA) supported Nix-TB study (a Phase 3 Study assessing the safety and efficacy of bedaquiline plus PA-824 plus linezolid in subjects with drug resistant pulmonary tuberculosis. https://clinicaltrials.gov/ct2/show/NCT02333799) for the treatment of pulmonary extensively drug-resistant forms of TB (XDR-TB—using a previous WHO definition of XDR-TB, which was MDR/RR-TB plus resistance to FQ and any of the SLI.) or treatment-intolerant or non-responsive MDR-TB and was conducted in sites in South Africa. At 6 months after the end of treatment, 11 patients (10%) had an unfavorable outcome and 98 patients (90%; 95% confidence interval (CI), 83–95) had a favorable outcome [5]. Patients with XDR-TB traditionally have had few treatment options and no standard treatment regimen. Published success rates for the treatment of XDR-TB were low and consistent across South Africa, averaging 14% and ranging from 2 to 22% [6, 7].

Based on the outcomes of the study, in August 2019, the US Food and Drug Administration, and in March 2020, the Committee for Medicinal Products for Human Use of the European Medicines Agency, recommended approval of Pa in combination with Bdq and Lzd for the treatment of pulmonary XDR-TB or treatment-intolerant or non-responsive MDR-TB.

In the June 2020 WHO guidelines, countries were recommended to replace the SLI with Bdq, and use of this all-oral Bdq-containing STR became the preferred treatment option [8]. The 2020 WHO guidelines also recommended the use of the Bdq, Pa and Lzd ("BPaL") regimen under operational research (OR) conditions for selected patients. The guidelines also stress the increased requirements for drug susceptibility testing (DST) and active TB drug safety monitoring and management (aDSM).

Here, we describe the processes, challenges, and interim results of the introduction of the BPaL regimen under OR conditions, via the provision of coordinated technical assistance (TA), in the 7 countries supported under the Korea International Cooperation Agency/TBA-funded "Leveraging Innovation for Faster Treatment of Tuberculosis (LIFT-TB)" project [9]. The primary objectives of the support were to firstly explore the feasibility of introducing the BPaL regimen, and to estimate its effectiveness and safety in MDR-/RR-TB patients with additional fluoroquinolone (FQ) resistance, and MDR-/RR-TB patients with documented treatment intolerance or failure. Secondly, through this support, the capacity of the respective NTPs was to be strengthened to implement the BPaL regimen, and the results of the OR activities were to be used to support the national scale-up of the BPaL regimen.

## 2. Processes for the introduction of BPaL regimen

The KNCV TB Foundation (hereafter referred to as "KNCV") had been working collaboratively with the TBA for several years on BPaL-related activities. During 2017–2019, acceptability, feasibility and costing studies were conducted in Indonesia, Kyrgyzstan, Nigeria on BPaL and BPaMZ regimens [10]. From 2019 to late 2020, KNCV supported the dissemination of the study results, the development of BPaL introduction plans/roadmaps in 4 countries (Indonesia, Kazakhstan, Uzbekistan, and Kyrgyzstan), helping to maintain the momentum for BPaL introduction in these countries. A draft generic BPaL OR protocol was developed and from October 2020, the introduction of BPaL was supported under the "LIFT-TB project" by a consortium comprised of TBA, KNCV and the International TB Research Center of South Korea (ITRC) in Indonesia, Kyrgyzstan, Ukraine (LIFT-TB support complementary to that provided under a TB REACH Wave 7 project), Uzbekistan, and Vietnam (KNCV-led) and in Myanmar and the Philippines (ITRC-led) [9].

### 2.1 Global support from KNCV

A generic BPaL OR protocol, a clinical guide, and standardized data collection tools (REDCap and Epi Info [with WHO EURO] electronic data collection platforms) were developed. In addition, generic BPaL-related site preparedness assessment tools were developed, and KNCV's generic DR-TB training package (developed under the earlier USAID-supported Challenge TB [CTB] Project) was updated. All such materials were shared with all countries for their adaptation and use. Technical assistance was provided to all countries with data collection and analyses. Drug procurement, especially in relation to Pa, was facilitated with various stakeholders (Global Drug Facility [GDF], Global Fund [GF], TBA, and Viatris [previously Mylan]). Lastly, a Global BPaL OR Coordination mechanism was established in late 2020 with multiple stakeholders, including WHO (HQ, AFRO and EURO), TDR, Stop TB Partnership (TB REACH), KNCV, TBA, and Viatris, South Africa's NTP and researchers, and ITRC. Twenty-one meetings were held up to February 2023 (when the mechanism was ended) between the partners to share experiences and challenges in their respective BPaL-related OR activities. In February 2023, a WHO convened "BPaLM Accelerator Platform" held its first meeting. The participants of the "Global BPaL OR Coordination mechanism" felt that this mechanism should be ended, and all subsequent efforts should be focused on supporting countries via the WHO convened "BPaLM Accelerator Platform".

KNCV's Generic BPaL OR Protocol was based initially on the Nix-TB trial conducted by the TBA, and updated accordingly after the Zenix-TB trial results became available [11–14]. The primary objectives were to estimate the effectiveness of the BPaL regimen by assessing the end-of-treatment outcome among patients treated with BPaL and to estimate the safety of the BPaL regimen by determining the rates of serious adverse events (SAE).

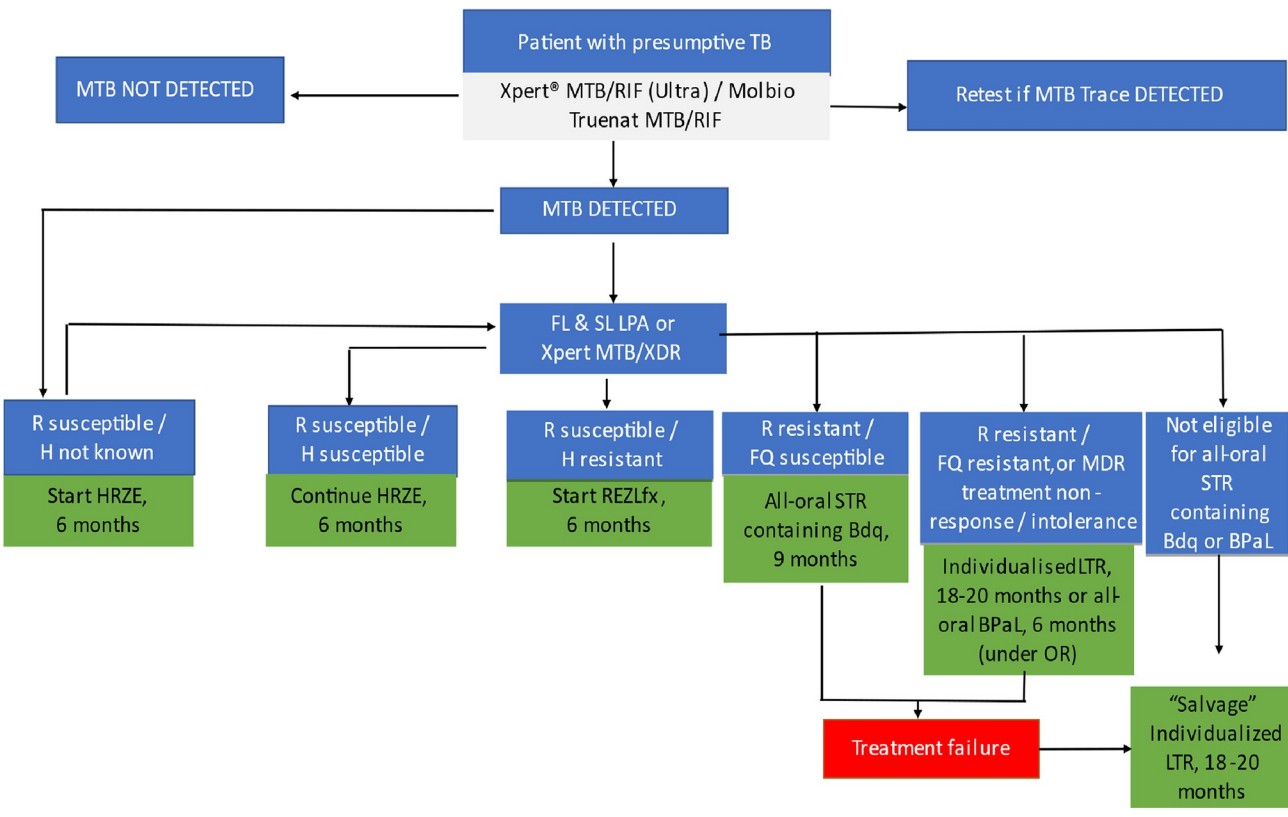

**Fig 1. Patient diagnostic pathway for the BPaL OR.**

Patients with bacteriologically confirmed RR-/MDR-TB with additional FQ resistance; or treated for RR-/MDR-TB, with documented non-response or intolerance to treatment, were eligible for BPaL (Fig 1).

Treatment was either hospital- or ambulatory-based according to the country regulations.

There was a standard treatment duration of 6 months (= 26 weeks). Treatment comprised of Bdq 400 mg once daily for 2 weeks, then 200 mg 3 times per week; Pa 200mg once daily; and Lzd 1200 mg once daily initially but adjustable according to clinical condition of the patient. In May 2022, WHO issued a "Rapid communication: Key changes to the treatment of DR-TB, which provided information to National TB Programmes on upcoming changes in WHO's recommendations regarding treatment of DR-TB. One change was to recommend Lzd 600mg from the start of the BPaL regimen and throughout treatment. In response, the generic BPaL OR protocol, V4 (Oct 2021) was updated to reflect this WHO recommendation regarding Lzd dose (V5, July 2022).

If the sputum culture remained positive after 4 months of treatment, duration could be extended by 3 months (i.e to a total of 39 weeks treatment). The protocol provided clear guidelines on how to modify the dose of Lzd if needed, how to manage temporary or permanent interruption of drugs or the full regimen, and when to discontinue the regimen. Safety monitoring was a crucial component of all the OR activities and was based on a clinical evaluation schedule. Building on existing safety monitoring systems in the respective countries, there was increased monitoring due to SAEs, adverse events (AEs) or co-morbidities. This required more intensive follow-up of the patient. If significant laboratory or ECG abnormalities were detected, even more frequent monitoring was to be done. There was a special focus on

monitoring for peripheral neuropathy, myelosuppression, optic neuritis and hepatoxicity as "adverse events of special interest (AESI)". A study evaluation form of AE was to be completed every visit. Patients were to be followed up at 6 months and 12 months post-treatment completion to assess the frequency of recurrent disease.

## 2.2 Country experiences

**Pre-implementation phase.** As was seen when introducing Bdq under the earlier CTB project, in-country advocacy was an important first step in countries. Especially in this case since the introduction of BPaL was to be done under OR conditions. Meetings of both in-country and KNCV HQ staff with the Ministries of Health (MoH) and the National TB Programmes (NTPs) were needed, as well as regular contact with local partners and other in-country stakeholders. Every forum was utilised to advocate for the introduction of BPaL. The OR activities were led by the respective NTP, with technical support from KNCV. It was important that this contributed to capacity building in the OR, laboratory strengthening, etc, and introduced a more effective patient-centered regimen for patients with highly DR-TB.

The various stakeholders were requested to be engaged in the OR activities either by contributing funding for the research (e.g for human resources, laboratory commodities, drugs), creating an implementing structure under the leadership of the NTP, project management (e.g by local NGOs), implementation of the OR activities, and participating in TB Expert Committees.

Plans for the BPaL OR were either developed or updated. The NTPs, with the support of the local KNCV teams or in-country teams, used the generic BPaL-related site preparedness assessment tools to assess the needs of health facilities and laboratories, patient care practices, and readiness. Depending on the current COVID-19-related restrictions, these assessments were done either in person or virtually. 'Pilot' sites were selected by the NTPs based on previous experience, availability of trained staff and infrastructure, etc.

With the engagement of the NTP, the National TB Reference Laboratory, national experts, and other partners (e.g WHO, GF, USAID, etc), the generic BPaL OR protocol was adapted to align with existing country programmatic management of drug-resistant TB (PMDT) implementation. There were multiple rounds of questions, technical sessions with KNCV TA consultants, and a "Frequently Asked Questions" document was developed (and incorporated into the generic BPaL OR Clinical Guide). After technical approval from the NTP, these national BPaL OR protocols were submitted to the relevant body for national ethics approval. The respective bodies were the: Ethics Committee of the Faculty of Medicine, University of Indonesia–Cipto, Mangunkusumo HospitaI, Indonesia; Ethics Committee, Ministry of Health, Bishkek, Kyrgyzstan; Institutional Review Board, Department of Medical Research, Yangon, Myanmar; Single Joint Research Ethics Board, Department of Health, Manila, Republic of the Philippines; Committee on Medical Ethics, National Institute of Phthisiatry and Pulmonology, Kyiv, Ukraine; Ethics Committee, Ministry of Health of the Republic of Uzbekistan, Tashkent, Uzbekistan; and Ethics Committee, Ministry of Health, Hanoi, Socialist Republic of Vietnam. In some countries, this was followed by MoH endorsement (including issuance of Memo to regions, etc.).

Although staff of several of the countries had had previous experience of introducing new drugs and regimens under the CTB project, a significant amount of capacity building was required. Hence BPaL-related "Training of Trainers" trainings for physicians and nurses, and members of the respective TB Expert Committees, were conducted. These were followed in the classic cascade manner by trainings of clinical and laboratory staff. National experts, staff of in-country KNCV and partners teams, and international staff from ITRC, KNCV and TBA acted as the trainers.

KNCV facilitated the drug procurement, especially of Pa, to the respective countries with the various stakeholders. Fifty patient courses of Pa were donated by Viatris to Kyrgyzstan. Pa was donated directly by the LIFT-TB project (via TBA) to Indonesia (100 courses), Myanmar (100), The Philippines (100), and Uzbekistan (50). Pa was procured under the TB REACH Wave 7 project for Ukraine (135), and with GF support for Vietnam (100).

Three electronic data collection platforms have been developed for the BPaL OR activities: REDCap (online web versions, real-time remote monitoring of the OR—used in Indonesia, Kyrgyzstan, The Philippines, Uzbekistan and Vietnam; Epi Info (offline, mobile/tablet/desktop versions, no real-time remote monitoring of the OR, instead data sharing regularly—used by Ukraine under the TB REACH Wave 7 project; and an Excel-based platform in Myanmar. Whilst maintaining the core set of variables agreed across the countries and the 6 developed standardized data collection forms, continuous refinement of the data collection tools was undertaken.

A standard set of 6 data collection forms was developed (1. Screening; 2. Enrolment; 3. Evaluation; 4. Treatment completion; 5. After treatment follow-up; and 6. Adverse events). Data collection is paper-based and digital, with data entry done either at the central level (e.g in Indonesia, Kyrgyzstan, Ukraine, Vietnam) or at the facility level (e.g in the Philippines). Screening data was entered into the electronic database irrespective of whether the patient was eligible for BPaL or not.

The BPaL OR data is owned by the country entity (NTP/MoH)–"data controller". KNCV provided TA to the countries in the data collection, analysis, and results dissemination–"data processor". KNCV processes the data as per the signed Data Sharing Agreement, aligning with the country's regulations and IRB requirements.

In 2021–2022, KNCV conducted multiple and varied on-line trainings of relevant national staff on the data collection process, the data collection forms, SOPs, and the REDCap database system in Indonesia, Kyrgyzstan, Myanmar, The Philippines, Uzbekistan, and Vietnam.

**Implementation phase.** The selection of patients to be screened for eligibility regarding the BPaL regimen was determined by the inclusion and exclusion criteria included in the respective country's OR protocol. All countries used the Generic OR protocol as their reference document and hence criteria for enrolment in the OR were very much standard across the countries. However, the characteristics of the patients enrolled on to BPaL were highly dependent on the epidemiology of the specific country. For example, in the Philippines, the FQ resistance amongst RR-/MDR-TB cases was low ($\approx$3%). As a result, twelve OR sites were selected, and initially most recruited patients were MDR-TB patients who were either intolerant or non-responsive to their treatment.

A schedule for monitoring clinical and bacteriological (smear and culture) progress of treatment, and blood and other tests for drug safety monitoring was developed by the respective country adapted from that in the generic BPaL OR protocol.

There was a requirement for BPaL drug powders for laboratory use, equipment or consumables in various countries tailored to the findings of laboratory need assessment and these were supported under the project using available funding.

Performing DST for Pa remains a challenging task. DST for Bdq and Lzd are routine phenotypic DSTs and the EQA for Bdq and Lzd became the part of annual EQA provided by respective SNRLs. However, no consensus on the Critical Concentration (CC) for Pa has been reached thus no recommendations on Pa DST are available as of the time of writing, hence no Pa EQA is available. To ensure a baseline QC for Pa DST for laboratories within LIFT TB which started performing Pa DST (LIFT TB protocol, MIC determination in 7 Pa dilution series), in 2022 the KNCV proposed the implementation of an external quality control procedure specifically for Pa. With the financial support of ITRC and technical support of Gauting

SNRL, the QA panel composed of a genotypically and phenotypically characterised panel of MTBC strains for Pa DST, was prepared and distributed to Kyrgyzstan, Ukraine, and Uzbekistan as an inter-laboratory QC procedure. The outcomes of this effort were successful, as the participating laboratories in the three countries passed the interlaboratory control for Pa phenotypic DST.

Coordination among partners and stakeholders was essential to harmonise the general approach and specific activities to maximize their impact and avoid duplication of effort. The progress in preparation for BPaL and implementation was closely monitored. Regular progress calls were held (every 2 weeks initially and then monthly with the LIFT-TB supported in-country teams) and monthly progress reports were submitted. The monitoring system established facilitated the identification of barriers to BPaL introduction, and targeted TA.

The first patient was enrolled into the BPaL OR cohort on 1 November 2020 in Ukraine, and the last patient on 31 March 2023 in Indonesia. All patients recruited into the BPaL OR cohort provided written informed consent for their inclusion in the OR.

**Challenges to implementation and with delivery of Pa and other agents.** There were numerous challenges to the implementation of the BPaL OR activities, including delivery of Pa and other agents. Several were resultant of the global COVID-19 pandemic. Mitigating actions were taken by the respective countries to overcome said challenges (Tables 1 and 2).

## 3. Interim results

Using Kyrgyzstan as the best-case example, it took 16–17 months from high level advocacy to enrolment of the first patient—of which 6 months were due to delays in drug delivery and customs release, with the first patients enrolled in August 2021 (Fig 2).

In the 7 countries supported by LIFT-TB project with their BPaL OR activities, 574 patients had been enrolled on BPaL by the end of April 2023 when enrolment under the LIFT-TB project ceased in all countries (Table 3). Most countries either enrolled the planned number of patients in the initial cohort or were close to it. The exception was Vietnam where the COVID-19 pandemic badly impacted the Health Services and the whole of society in the country. All study sites for the screening and enrolment of both DS-TB and DR-TB patients were affected. Regarding screening, there were difficulties in sputum transportation for Xpert and SL LPA (the laboratory staff were overloaded with COVID-19 testing and with COVID-19 illness amongst themselves). Also, patients were reluctant to go to health care facilities due to the fear of COVID-19, the requirement of COVID-19 screening tests, and the difficulties in moving / travelling around during the COVID-19 lockdown period. For enrolment, TB hospitals / facilities and staff were fully or partially redeployed to the COVID-19 response effort (treatment, vaccine, screening), difficulty in requesting the initial inpatient treatment (1 month), and staff themselves falling ill with COVID-19. Many of the restrictions imposed due to COVID-19 remained in place in Vietnam well into 2022. The first in-person monitoring visits took place only in May 2022. It also took many months of 2022 for the TB services to revert to providing TB services. By then, the issue of enrolment of patients to a competing OR activity had arisen. Additional constraints in Vietnam were competition with another research trial recruiting the same group of patients, initial requirement for hospital admission of all BPaL patients, and enforcement of new Good Clinical Practice regulations).

For those patients who have a final treatment outcome (as of January 2024), the treatment success rate is extremely encouraging at 90.9% (280 / 308) (Table 3). However, this is only an interim set of results from the REDCap database (N = 308) and does not currently include data from other sources for Myanmar and Ukraine (N = 253). Results are consistent across countries, with a range of 86% (Kyrgyzstan) to 95–97% (Indonesia, the Philippines and Vietnam).

**Table 1. Challenges to implementation.**

| Challenges to implementation | Mitigating actions taken |
| --- | --- |
| 1. Overall project was delayed by 6 months and the final budget was significantly reduced due to the global COVID-19 pandemic. | 1. Certain activities (e.g separate trainings for laboratory staff, engagement of community organizations) included in the initial plans were shelved. |
| 2. Meetings and assessments were delayed due to late start of project. And due to the global COVID-19 pandemic, face-to-face site assessments, trainings and project monitoring were not possible, leading to delays. | 2. Remote/virtual means were maximized for communication, TA, and training. Physical visits to the field were re-started as soon as possible. In the interim, remote/virtual assessment(s) was conducted. |
| 3. Long in-country OR protocol approval and implementation processes. | 3. TA and inputs from all levels (NTP, NTRL, etc) were provided, and local authorities/counterparts engaged, to expedite approvals of OR- related documents. |
| 4. Stringent legal requirements in data sharing in some countries. | 4. Memo of Agreements, Data Sharing Agreements and Non-Disclosure Agreements were all developed and signed with the respective MOHs to ensure the smooth sharing of data. |
| 5. Issues with human resources and facilities, including many TB Health Care Workers (HCW), facilities and laboratories being repurposed for the COVID-19 response in the countries; HCW falling ill with COVID or being quarantined; DST capacity for Bdq, Dlm, Lzd remained limited and non-existent for Pa; provision of back-up freezing of isolates a challenge; and specimen transportation problems still existed in several countries. | 5. To overcome issues with facilities and transportation, a supply of Bdq, Lzd and Pa drug powder for DST and linked DST training for laboratory staff was done by ITRC. ITRC also provided a limited amount of equipment to countries (e.g provision of back-up freezer to Uzbekistan), and in the Philippines specimen transport guidelines and meetings with a courier to ensure cold chain all through transportation. |
| 6. There was a lower number of patients enrolled than was anticipated, due to a significant decline in TB notifications in general because of the COVID-19 pandemic, patients having prior exposure to Bdq and/or Lzd for more than 4 weeks, and difficult clinical cases being deemed ineligible for BPaL. | 6. To boost enrolment numbers, patient eligibility was widened. Patients from other states/provinces, were included, with travel support; BPaL OR pilot sites expanded to other states/provinces; active case finding in highest DR-TB burden areas, including contact investigation, and inclusion of patients who were shown to be Bdq and/or Lzd sensitive by DST irrespective of their prior drug exposure duration. To improve the management of enrolled patients, video DOT was introduced in certain countries, an International Expert Committee (with members from KNCV and TB Alliance) was created and involvement of WHO EURO in the European Region, to support and strengthen patient management. |
| 7. Limited face-to-face follow-up visits may have led to delayed detection and management of AEs. | 7. Face to face visits to patients were re-started as soon as it was possible. |
| 8. Equipment and consumables | 8. In the initial stages, a local procurement approach was adopted, but later the approach was streamlined to the GDF procurement process, considering the variations in procedures and regulations across the different countries. |
| 9. Training to Ukraine | Due to the war conflict and resultant inability to access Ukraine, ITRC collaborated with Gauting Laboratory in Germany, to provide various technical assistance and education initiatives in the region. |

Although a significant number of patients had an adverse event of special interest (AESI) (from REDCap database, 235 AESIs were reported), these were well managed, and although a significant percentage of patients had to decrease the dose or stop Lzd, only 1 patient had to discontinue the full BPaL regimen (Table 4). In addition, no unexpected SAEs or AEs were observed.

## 4. Discussion

In recent years, there have been major improvements in the treatment available for MDR-/ RR-TB patients. From the pre-2018 WHO recommended conventional MDR-TB regimen

**Table 2. Challenges to delivery of Pa and other agents.**

| Challenges to delivery of Pa and other agents | Mitigating actions taken |
|---|---|
| 1. Due to the COVID-19 pandemic, travel restrictions were imposed on transportation systems (including the closure of airports), exportation and importation regulations were tightened, and staff were redeployed or fell ill. These all contributed to long delays in drug deliveries. | 1. Constant communication with local agencies with assistance from NTPs and local drug distributors, and international partners (i.e GDF, KNCV, TBA and Viatris), led to delays to drug deliveries being minimised. |
| 2. Multiple reviews of protocols and extensive documentation required by Viatris prior to any Pa donation also led to delays in drug delivery. | 2. Switching of donation from Viatris to directly from the LIFT-TB project sped up the delivery of Pa to *select* countries. |
| Multiple challenges were identified in TB laboratories during the OR, although acknowledging that not every laboratory faced the same level of challenges. These issues included: contamination in sample collection; transportation-related challenges; stockouts of essential test kits; and the risk of leaving TB samples unattended due to excessive workload. Moreover, even when tests were conducted, some results could be unreliable, and delays occurred in result reporting and data encoding, subsequently affecting timely communication with the requesting sites. Infrastructure issues, including interruptions in water and electricity supply, and biosafety concerns were also noted, along with the challenge of maintaining an adequate workforce. | |

(18–20 months treatment, with an 8-months intensive phase including a daily injection), to a 9–12 months STR but still with a SLI agent included, and then from June 2020 onwards, an all-oral 9-months STR with Bdq replacing the SLI and the 6-months all-oral BPaL regimen. However, the latter regimen was recommended only for use under OR conditions. Yet again, the 2020 guidelines stressed the increased requirements for DST and aDSM.

Although the process of introducing BPaL under OR was thought to be lengthy, due to earlier preparation under CTB in several countries for introducing Bdq, Dlm and the STR, the timelines were in fact shorter than the average of 2 years observed for the introduction of Bdq under the CTB project [15]. Implementation of BPaL under OR conditions was shown to be feasible in a variety of challenging settings. And as has been observed by others when Bdq and the STR were introduced, with careful planning and monitoring, decentralised implementation was shown to be feasible in various settings [16].

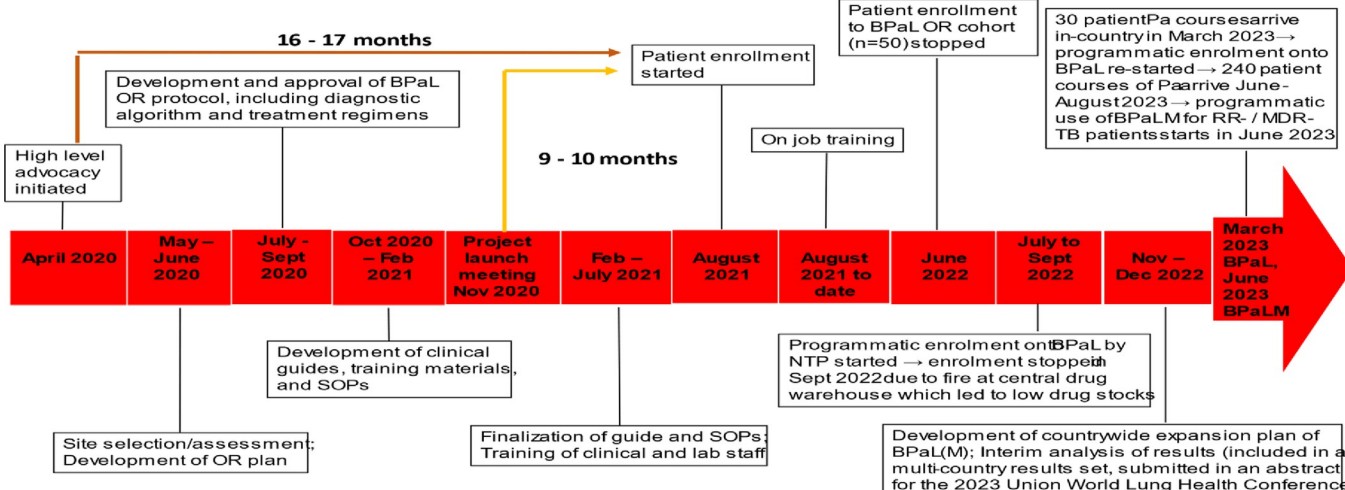

**Fig 2. Timeline of BPaL OR and programmatic introduction of BPaL-based regimens in Kyrgyzstan, 2020–2023.**

**Table 3. Patient enrolment and end of treatment success results (from *interim* analyses of REDCap database), under the BPaL OR component of the LIFT-TB project (REDCap database version of 29 January 2024).**

| Country (Project) | Nos of pts planned in 1st cohort | Start of pt enrolment | Nos of pts enrolled | End of pt enrolment under LIFT-TB OR | Late Screen failures [a] | Nos of pts with end of treatment outcome evaluation in REDCap | End of treatment success (%) [b] |
|---|---|---|---|---|---|---|---|
| Indonesia (LIFT-TB) | 100 | July 2022 | 87 | End March 2023 | 1 | 86 | 82 / 86 (95.3%) |
| Kyrgyzstan (LIFT-TB) | 50 | Aug 2021 | 50 | End June 2022 | 1 | 50 | 43 / 50 (86%) |
| Myanmar (LIFT-TB) | 100 | Dec 2021 | 100 | End March 2023 | Not Applicable | Not Applicable | |
| Philippines (LIFT-TB) | 99 | May 2021 | 99 | End Dec 2022 | 0 | 93 | 82 / 84 (97.6%) |
| Ukraine (TB REACH W7/ LIFT-TB) | 135 | Nov 2020 | 153 | TB REACH ended Dec 2021 | Not Applicable | Not Applicable | |
| Uzbekistan (LIFT-TB) | 50 | Jan 2022 | 43 | End Dec 2022 | 2 | 39 | 34 / 39 (87.2%) |
| Vietnam (LIFT-TB) | 100 | Nov 2021 | 40 | End Dec 2022 | 0 | 40 | 38 / 40 (95%) |
| TOTAL | 638 | | 574 | | 4 | 308 | 280 / 308 (90.9%) |

[a] "Late Screen failures" refers to identifying participants who were initially enrolled in the study but were later found to be ineligible due to not meeting the inclusion/ exclusion criteria. This determination is made based on information not initially available during the screening process. For instance, late screen failures may include individuals who exhibit resistance to Bdq or Lzd, as detected in the baseline culture samples.

[b] Modified intention to treat analysis (MITT), patients withdrawn from the BPaL treatment due to baseline resistance to any of the BPaL component drugs are not included in the denominator of the treatment success. However, this may not be true for all countries on these analyses. Interim results are from the data in the REDCap database for Indonesia, Kyrgyzstan, The Philippines, Uzbekistan, and Vietnam (N = 308).

Crucial to this success were: initial intensive advocacy and frequent communication with the different levels of the national authorities and engagement of regulatory bodies; detailed situational assessment and planning conducted; and activities linked to further strengthening of the health systems prior to initiation of patients on treatment. Also, the engagement and collaboration with both local and international partners was highly important. There were many challenges posed by the global COVID-19 pandemic, however these were often mitigated by maximizing available solutions (Tables 1 and 2).

The introduction of BPaL under OR conditions was a novel experience in many of the countries, and initially clinical experience was limited. However, as experience grew, so too did confidence, particularly as patients who had been previously thought to be "incurable" were cured. This became an important advocacy tool to enhance the prospect of introducing new medicines and regimens, with countries benefiting from sharing of experiences between

**Table 4. BPaL safety data under BPaL OR: Indonesia, Kyrgyzstan, The Philippines, Uzbekistan, Vietnam (N = 323, REDCap database version from 29 January 2024).**

| Adverse Event Type | Nos of patients with ≥1 AELTM / AESI / SAE (%) |
|---|---|
| Adverse Event(s) Leading to Treatment Modification (AELTM) | 148 (46%) |
| Adverse Events of Special Interest (AESI) [3] | 235 (73%) |
| Serious Adverse Events (SAE) | 79 (24%) |

[3] Included myelosuppression, peripheral neuropathy, optic neuritis, hepatotoxicity, and $QT_c$ prolongation.

themselves. Guidelines and protocols were constantly being updated to align with the latest available scientific evidence.

The interim treatment outcomes show excellent results and are very much in line with the findings of the Nix-TB and Zenix-TB trials [5, 13]. Because of the worrisome known AEs, especially those linked with Lzd, it was of crucial importance that functioning aDSM systems were in place [17]. Although a significant number of patients had an AESI, these were well managed and few patients had to discontinue the full regimen. Again, these findings are very much in line with earlier findings of the Nix-TB and Zenix-TB trials [5, 13]. In addition, no unexpected SAEs or AEs were observed. Further analyses will be done to see whether the reduction of the initial dose of Lzd from 1200mg to 600mg daily, was matched with a reduction in AEs/ AESIs attributable to the use of Lzd. The OR results support the use of BPaL in patients classed as eligible as per the OR protocol [14].

Further significant advances in the treatment of DR-TB patients, with new WHO recommendations issued in late 2022, have occurred during the implementation period of the BPaL OR activities reported here. New evidence on DR-TB treatment has become available to WHO after the issuance of its latest Consolidated Guidelines in June 2020.[7] WHO convened an independent Guideline Development Group (GDG) in February–March 2022 to assess the results of analyses of the new data, using the international Grading of Recommendations Assessment, Development and Evaluation (GRADE) approach to the assessment of scientific evidence, which were presented to the GDG during its meetings [18].

Several new regimens were analysed, including a new 6-month regimen based on BPaL in combination with moxifloxacin (BPaLM), which was evaluated in the TB-PRACTECAL randomized clinical trial in which a novel shorter, all-oral treatment regimen for MDR-/RR-TB or pre-XDR-TB was tested [19]; the 6-month regimens based on the BPaL combination with decreased exposure to Lzd (lower dosing or shorter duration) evaluated in the ZeNix study [12, 13]; and the modified all-oral shorter regimens (6–9 months or 9–12 months of Bdq-Lzd-Lfx-Z-Eto/Hh/Trd,) containing all three Group A medicines evaluated in the NeXT trial for the treatment of MDR-/RR-TB without added FQ resistance [20] or implemented by the NTP in South Africa. Following the GDG meeting, the WHO released a "Rapid communication: Key changes to the treatment of drug-resistant tuberculosis" in May 2022, which aimed to inform NTPs and other stakeholders about the key implications for the treatment of DR-TB, to allow for a rapid transition and planning at country level [21]. Subsequently in December 2022, WHO issued an updated DR-TB treatment Guideline and Operational Handbook [22, 23].

The two key changes are:

- The 6-month BPaLM regimen, comprising Bdq, Pa, Lzd (600 mg) and moxifloxacin (Mfx), may be used programmatically in place of 9-month or longer (>18 months) regimens, in patients (aged 14 years and older) with MDR-/RR-TB who have not had previous exposure to Bdq, Pa and Lzd (defined as greater than 1 month exposure). This regimen may be used without Mfx (i.e BPaL) in the case of documented resistance to FQs (i.e in pre-XDR-TB patients).

- The 9-month, all-oral, Bdq-containing regimens (4–6 Bdq[6]-Lfx[Mfx]-Lzd[2]-E-Z-Hh-Cfz / 5 Lfx[Mfx]-Cfz-Z-E, with 2 months of Lzd replacing 4 months of Eto as used in South Africa since 2018, or 4–6 Bdq[6]-Lfx[Mfx]-Eto-E-Z-Hh-Cfz / 5 Lfx[Mfx]-Cfz-Z-E) are preferred over the longer (>18 months) regimen in adults and children with MDR-/RR-TB, without previous exposure to second-line treatment (including Bdq), without FQ resistance and with no extensive pulmonary TB disease or severe extrapulmonary TB. In these regimens, 2 months of Lzd (600 mg) can be used as an alternative to 4 months of ethionamide.

Access to rapid DST for ruling out FQ resistance is required before starting a patient on one of these regimens.

These changes provide much optimism for the prospects of both DR-TB patients and NTPs. However, challenges remain around the use of new drugs (e.g Bdq) and repurposed drugs (e.g Lzd) [24, 25]. Hopes are also tempered by the fear of emerging resistance, which reinforces the critical need for access to rapid and reliable DST for all drugs, including Bdq, Pa, Dlm and repurposed drugs (FQs, Lzd and Cfz) [26, 27].

However, with the release of updated guidance on DR-TB treatment from the WHO, the TB world is moving into an era of being able to offer shorter effective all-oral treatments to most DR-TB patients. Already the potential of safer shorter effective all-oral treatments is being glimpsed [28]. The LIFT-TB Project's main support in 2023 and 2024 aims at assisting countries to introduce the BPaL-based regimens as recommended by WHO for a much wider group of DR-TB patients and under programmatic conditions as soon as possible, whilst wrapping up the BPaL OR reported on here.

## 5. Conclusion

The experiences of the LIFT-TB Project show that, with careful advocacy, frequent communication with local and international partners, and work to strengthen health systems, BPaL OR for pre-XDR-TB patients was feasible regardless of the different country settings. However, it is important to follow a series of steps to address essential aspects of the delivery system. Although the process appeared slow, due to earlier health system strengthening work done by the CTB project, the BPaL OR process went quicker than the introduction of Bdq and Dlm under CTB. Many challenges were due to the COVID-19 pandemic, but these were mitigated by maximizing available solutions.

There was constant updating of guidelines and other project documents to align with the latest scientific evidence. As experience grew, countries were motivated by their own and other countries' success–the sharing of information, experiences, and interim results had a significant positive effect within and across countries. Interim OR results show excellent patient responses to treatment and are comparable to those seen in the Nix-TB, ZeNix-TB, and TB PRACTECAL studies. SAEs were limited, AEs and AESIs although common were manageable, and no unexpected AEs were observed.

With the release of the 2022 updated guidance on DR-TB treatment from the WHO, the LIFT-TB Project's main support in 2023 and 2024 aims at assisting countries programmatically introduce the BPaL-based regimens for a much wider group of DR-TB patients as soon as possible, whilst wrapping up the BPaL OR reported on here. Targeted technical assistance and advocacy will be critical to accelerate uptake and transitioning to the best available treatment regimens.

## Supporting information

**S1 File. Uzbekistan ERB approval translation with PI named.**
(DOCX)

**S2 File. Kyrgyzstan ERB approval translation with PI named.**
(DOCX)

**S3 File. ERB approvals.**
(ZIP)

## Acknowledgments

The authors thank all the leadership and DR-TB teams of the NTPs in Indonesia, Kyrgyzstan, Myanmar, Philippines, Ukraine, Uzbekistan, and Vietnam, the health care workers diagnosing and caring for the patients and the staff in the in-country teams in the respective countries, for all their hard work and patience. And thanks to Naoko Doi and Caroline Marck-Haehnel for their work and assistance during their time as consultants with TBA. All activities reported on were funded under the auspices of the KOICA/TBA-funded LIFT-TB project (Indonesia, Kyrgyzstan, Myanmar, Philippines, Ukraine, Uzbekistan, and Vietnam), and the TB REACH Wave 7 project in Ukraine. The authors wish to thank the South Korean people for their generous support through KOICA.

## Author Contributions

**Conceptualization:** D. F. Wares, M. Mbenga, V. Mirtskhulava, S. N. Cho, U. Go, J. S. Lee, D. Everitt, S. Juneja, A. Gebhard.

**Data curation:** V. Mirtskhulava.

**Formal analysis:** V. Mirtskhulava.

**Funding acquisition:** D. F. Wares, S. N. Cho, U. Go, J.-K. Jung, M. Diachenko, S. Juneja, A. Gebhard.

**Investigation:** D. F. Wares, M. Mbenga, V. Mirtskhulava, M. Quelapio, A. Slyzkyi, I. Koppelaar, S. N. Cho, U. Go, J. S. Lee, D. Everitt, S. Foraida, M. Diachenko, S. Juneja, E. Burhan, A. Totkogonova, Z. Myint, I. Flores, N. A. Lytvynenko, N. Parpieva, N. V. Nhung, A. Gebhard.

**Methodology:** D. F. Wares, V. Mirtskhulava, M. Quelapio, S. N. Cho, U. Go, D. Everitt, S. Foraida, A. Gebhard.

**Project administration:** D. F. Wares, M. Mbenga, S. N. Cho, U. Go, J.-K. Jung, M. Diachenko, S. Juneja, E. Burhan, A. Totkogonova, Z. Myint, I. Flores, N. A. Lytvynenko, N. Parpieva, N. V. Nhung, A. Gebhard.

**Resources:** J. S. Lee, J.-K. Jung.

**Software:** V. Mirtskhulava.

**Supervision:** D. F. Wares, M. Mbenga, V. Mirtskhulava, M. Quelapio, A. Slyzkyi, I. Koppelaar, J.-K. Jung, D. Everitt, M. Diachenko, S. Juneja, E. Burhan, A. Totkogonova, Z. Myint, I. Flores, N. A. Lytvynenko, N. Parpieva, N. V. Nhung, A. Gebhard.

**Validation:** V. Mirtskhulava.

**Visualization:** D. F. Wares, M. Quelapio, S. N. Cho.

**Writing – original draft:** D. F. Wares, M. Mbenga, V. Mirtskhulava, M. Quelapio, A. Slyzkyi, I. Koppelaar, S. N. Cho, U. Go, J. S. Lee, J.-K. Jung, D. Everitt, S. Foraida, M. Diachenko, S. Juneja, E. Burhan, A. Totkogonova, Z. Myint, I. Flores, N. A. Lytvynenko, N. Parpieva, N. V. Nhung, A. Gebhard.

**Writing – review & editing:** D. F. Wares, M. Mbenga, V. Mirtskhulava, M. Quelapio, A. Slyzkyi, I. Koppelaar, S. N. Cho, U. Go, J. S. Lee, J.-K. Jung, D. Everitt, S. Foraida, M. Diachenko, S. Juneja, E. Burhan, A. Totkogonova, Z. Myint, I. Flores, N. A. Lytvynenko, N. Parpieva, N. V. Nhung, A. Gebhard.

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
