## [Decision Letter · Decision Letter 0]

21 Jun 2024

PONE-D-24-06032Introducing BPaL: experiences from countries supported under the LIFT-TB ProjectPLOS ONE

Dear Dr. Wares,

Thank you for submitting your manuscript to PLOS ONE. After careful consideration, we feel that it has merit but does not fully meet PLOS ONE’s publication criteria as it currently stands. Therefore, we invite you to submit a revised version of the manuscript that addresses the points raised during the review process.

Thank you for submitting your well written manuscript to this journal. We believe that the comments from the reviewers are very constructive, should be fairly easy to respond to, and will improve the overall impact of the manuscript. We look forward to receiving the revised manuscript soon. 

We look forward to receiving your revised manuscript.

Kind regards,

Shahriar Ahmed, MBBS, MHE, MPhil

Academic Editor

PLOS ONE

Journal Requirements:

2. Thank you for stating in your Funding Statement: "Activities under the “Leveraging Innovation for Faster Treatment of Tuberculosis (LIFT-TB)” project in the seven countries and at KNCV TB Foundation, the Netherlands and the International TB Research Center, Republic of Korea, global levels, were supported by funding from the Korea International Cooperation Agency (KOICA), Republic of Korea and the TB Alliance (TBA), the United States of America. The following authors received partial salary support via the LIFT-TB project: DFW; MM; VM; MQ; AS; IK; SNC; UG; JSL; J-KJ; and AG. KOICA had no role in the operational research study design, data collection and analysis, decision to publish, or preparation of the manuscript. As lead co-ordinating partner in the LIFT-TB project, TBA did play a role in the operational research study design, data collection and analysis, decision to publish, and preparation of the manuscript.  

KOICA https://www.koica.go.kr/ TBA. TB Alliance | Putting science to work for better, faster TB cures."

3. Thank you for stating the following in the Competing Interests section: "I have read the journal's policy and the following co-authors of this manuscript -S. Foraida, M. Diachenko, and S. Juneja have the following competing interests: Paid employment or consultancy with TB Alliance, the developer of pretomanid."

5. Please ensure that you include a title page within your main document. You should list all authors and all affiliations as per our author instructions and clearly indicate the corresponding author.

7. Please include captions for your Supporting Information files at the end of your manuscript, and update any in-text citations to match accordingly. Please see our Supporting Information guidelines for more information: 

Additional Editor Comments:

Thank you for submitting your well written manuscript to this journal. We believe that the comments from the reviewers are very constructive, should be fairly easy to respond to, and will improve the overall impact of the manuscript. We look forward to receiving the revised manuscript soon.

Reviewers' comments:

Reviewer's Responses to Questions

**Comments to the Author**

1. Is the manuscript technically sound, and do the data support the conclusions?

Reviewer #1: Yes

Reviewer #2: Yes

2. Has the statistical analysis been performed appropriately and rigorously? 

Reviewer #1: N/A

Reviewer #2: N/A

3. Have the authors made all data underlying the findings in their manuscript fully available?

Reviewer #1: No

Reviewer #2: No

4. Is the manuscript presented in an intelligible fashion and written in standard English?

Reviewer #1: Yes

Reviewer #2: Yes

5. Review Comments to the Author

Reviewer #1: Thanks to the authors for their research paper titled “Introducing BPaL: experiences from countries under the LIFT-TB Project” which describes the processes, the challenges, and interim results of the introduction of the BPaL regimen under OR conditions in 7 countries. This was a timely step taken under operational research conditions to test/validate the study results observed in a previous study named Nix-TB study. The authors have presented the processes, challenges and interim results of their study very well.

The manuscript is technically sound and the data supports the conclusions.

Despite their efforts the authors couldn’t make all data underlying in the findings their manuscript fully available as the data cannot be shared publicly because of legal ownership restrictions. All data is the property of the respective countries. The authors have access to the respective database via data sharing agreements signed with the MOH of each country and the data underlying the results can be made available if permission is granted by the respective MOH for researchers who meet the criteria for access to confidential data.

The authors may consider the below suggestions for their paper:

Background: The WHO recommendation on shorter 9-month treatment regimen (STR) was made in 2016 (https://www.who.int/news/item/12-05-2016-rapid-diagnostic-test-and-shorter-cheaper-treatment-signal-new-hope-for-multidrug-resistant-tuberculosis-patients) but in the background (last sentence) it is mentioned - “Since 2018, the WHO has recommended that for certain MDR/RR-TB.......”

Table 3: Please check the figures and their totals for correctness e.g. there are differences in Philippines between the “number of patients with end of treatment outcome evaluation in REDCap (93)” and “End of treatment success (84)” – is there is any reason why only 84 were evaluated for treatment success while 93 had end of treatment outcome data? Please check the totals (at the bottom row [(82+43+82+34+40) / (86+50+84+39+40) = 279/299 = 93.3%] of the last column with “End of treatment success (%)” which is shown 280/308 (90.9%)

Table 4: BPaL safety data showing the numbers with adverse event types but not presenting the numbers requiring decreasing the dose or stop of LZD and the number of patients discontinued treatment. Please specify what types of treatment modifications (dose of Lzd decreased or Lzd stopped or full treatment regimen discontinued?) accounted under “adverse events leading to treatment modification (AELTM)”

Discussion (the two key changes are): recommended age of patients for the 6-month BPaLM regimen is 14 years and above (it is mentioned in the manuscript ≥15 years) and previous exposure to Bdq, Pa and Lzd defined as 1 month or more (in the manuscript >1-month exposure).

Reviewer #2: The manuscript reports on "Introducing BPaL: experiences from countries supported under the LIFT-TB Project". It is well written, clear and comprehensive. The experience from the BPaL operational research - including on the processes, lessons learned, challenges, mitigation measures implemented and the interim results - are all important to NTPs, in-country and international stakeholders. The interim results (excellent treatment outcome) could contribute to building confidence and improve uptake of the 6-month regimen being implemented in programmatic conditions.

I agree and appreciate the research team's decision to "assisting countries programmatically introduce the

BPaL-based regimens for a much wider group of DR-TB patients as soon as possible, whilst wrapping

up the BPaL OR reported on here in line with the WHO's revised recommendation/guidance in 2023 and 2024". Targeted technical assistance and advocacy would be critical to accelerate uptake and transitioning to the best available treatment regimens.

Minor comments

- I would suggest revising the term "prove" as OR mayn't be the best option to do so (abstract and 5th para - background).

- Page 11 1st para, line 3 (Background) - delete "each year" as the year is already specified (2022) in the same sentence.

- Provide the data on the number of patients who discontinued treatment due to SAEs rather than saying "few" (abstract, and Result page 22). This information (if available) could be added to Table 4 (page 22) as well.

- Please clarify why the interim results from Myanmar and Ukraine were not available in Q2/2024 (253 patients? is this related to the contexts in the countries? do you anticipate different outcome?

- In Vietnam, was it due to "very prolonged impact of the COVID-19 pandemic..." or late initiation of the pandemic (unlike in most other countries, the pandemic started in Vietnam in 2021)?

6. PLOS authors have the option to publish the peer review history of their article (what does this mean?). If published, this will include your full peer review and any attached files.

Reviewer #1: No

Reviewer #2: **Yes: **Mohammed Yassin (MD, MSc, PhD),

---

## [Author Response · Author response to Decision Letter 0]

30 Jul 2024

Reviewer #1

The authors may consider the below suggestions for their paper:

Background: The WHO recommendation on shorter 9-month treatment regimen (STR) was made in 2016 (https://www.who.int/news/item/12-05-2016-rapid-diagnostic-test-and-shorter-cheaper-treatment-signal-new-hope-for-multidrug-resistant-tuberculosis-patients) but in the background (last sentence) it is mentioned - “Since 2018, the WHO has recommended that for certain MDR/RR-TB.......”

Response: Thanks to the reviewer for this correction. The text has been amended and now reads as “Since 2016, the WHO has recommended that for certain MDR-/RR-TB patients, a shorter 9-12 months treatment regimen (STR), with a SLI agent included, could be used instead of a longer (preferably all-oral) regimen.3 From 2018 onwards, WHO has considered that a 9-months STR with bedaquiline (Bdq) replacing the SLI agent could be used.4” A new reference 3 has been added and all the subsequent previous references have been renumbered. 

Table 3: Please check the figures and their totals for correctness e.g. there are differences in Philippines between the “number of patients with end of treatment outcome evaluation in REDCap (93)” and “End of treatment success (84)” – is there is any reason why only 84 were evaluated for treatment success while 93 had end of treatment outcome data? Please check the totals (at the bottom row [(82+43+82+34+40) / (86+50+84+39+40) = 279/299 = 93.3%] of the last column with “End of treatment success (%)” which is shown 280/308 (90.9%)

Response: Differences between the numbers for the number of patients enrolled, end of treatment outcome evaluation in REDCap and end of treatment success, is due to the time factor and exclusions. The data presented is from the ‘REDCap database on 29 January 2024’. This is a dynamic process, with data cleaning, validation and analyses, and numbers have changed since then due to data being added to the database. And although 99 were known to have been enrolled into the BPaL OR patient cohort at that date, this number can change due to “Late Screen failures” as shown in Column 6 of Table 3 and defined in footnote 1. Further changes in the denominator totals in Column 8 “End of treatment success” can occur due to it showing the results of a “ Modified intention to treat (MITT) analysis” with exclusions being made (please refer to footnote 2 for further details). These factors also apply to the totals shown in the bottom row. To stress the dynamic on-going process of the project’s data analyses, the term “interim” has been added to the title “Table 3. Patient enrolment and end of treatment success results (from interim analyses of REDCap database), under the BPaL OR component of the LIFT-TB project (REDCap database version of 29 January 2024)”

Table 4: BPaL safety data showing the numbers with adverse event types but not presenting the numbers requiring decreasing the dose or stop of LZD and the number of patients discontinued treatment. Please specify what types of treatment modifications (dose of Lzd decreased or Lzd stopped or full treatment regimen discontinued?) accounted under “adverse events leading to treatment modification (AELTM)”

Response: To address this point would almost require a separate manuscript. A multi-country manuscript, including data from all 7 countries, detailing out effectiveness and safety data in detail, is currently being prepared for submission to an international peer-reviewed scientific journal. 

Discussion (the two key changes are): recommended age of patients for the 6-month BPaLM regimen is 14 years and above (it is mentioned in the manuscript ≥15 years) and previous exposure to Bdq, Pa and Lzd defined as 1 month or more (in the manuscript >1-month exposure).

Response: Text amended accordingly

Reviewer #2 

Minor comments

- I would suggest revising the term "prove" as OR mayn't be the best option to do so (abstract and 5th para - background).

Response: Text in abstract and 5th paragraph - background has been amended in line with suggestion of Reviewer #2. Now reads in the abstract as “The OR objectives were to explore the feasibility of introducing the BPaL regimen,… ” and 5th paragraph – background as “The primary objectives of the support were to firstly explore the feasibility of introducing the BPaL regimen,…”.

- Page 11 1st para, line 3 (Background) - delete "each year" as the year is already specified (2022) in the same sentence.

Response: Text amended.

- Provide the data on the number of patients who discontinued treatment due to SAEs rather than saying "few" (abstract, and Result page 22). This information (if available) could be added to Table 4 (page 22) as well.

Response: Text amended. Now reads in the abstract as “… and only 1 patient had to discontinue the complete BPaL treatment regimen.” and in ‘Results, page 22’ as “…. only 1 patient had to discontinue the full BPaL regimen…. As it was just the one patient, the information has been added in the text but not felt necessary to add in Tabe 4 as well. 

- Please clarify why the interim results from Myanmar and Ukraine were not available in Q2/2024 (253 patients? is this related to the contexts in the countries? do you anticipate different outcome?

Response: Myanmar and Ukraine did not use the REDCap database, choosing instead to use an Excel-based database and Epi Info database respectively. The data from these databases were not available as of the time of writing of the manuscript. And when available, required incorporation with the data from the REDCap database, resulting in a much longer time for cleaning, validation and analyses of the data. On current evidence, we do not anticipate any differences in the outcomes from those already reported in this manuscript. A multi-country manuscript, including data from all 7 countries, detailing out effectiveness and safety data in detail, is currently being prepared for submission to an international peer-reviewed scientific journal.

- In Vietnam, was it due to "very prolonged impact of the COVID-19 pandemic..." or late initiation of the pandemic (unlike in most other countries, the pandemic started in Vietnam in 2021)?

Response: Text amended to clarify the prolonged COVID impact in Vietnam. The main factor was that the responses to the COVID pandemic introduced by the Government in Vietnam (e.g. redeployment of TB staff and TB hospitals to the COVID response were in place for a much longer time than in other countries). Now reads in the Interim Results section, second paragraph as “The exception was Vietnam (due to a very prolonged impact of the COVID-19 pandemic mainly due to the responses to the COVID pandemic introduced by the Government [e.g. redeployment of TB staff and TB hospitals to the COVID response] were in place for a much longer time than in other countries),…”.

All Editorial and Journal Requirements have been adhered to in amended and revised documents that have been uploaded.

---

## [Decision Letter · Decision Letter 1]

4 Sep 2024

PONE-D-24-06032R1Introducing BPaL: experiences from countries supported under the LIFT-TB ProjectPLOS ONE

Dear Dr. Wares,

Thank you for submitting your manuscript to PLOS ONE. After careful consideration, we feel that it has merit but does not fully meet PLOS ONE’s publication criteria as it currently stands. Therefore, we invite you to submit a revised version of the manuscript that addresses the points raised during the review process.

I would like to congratulate the authors for this very well written manuscript and timely submission of the revised version. There is just one minor comment left to be addressed from reviewer 2. Please address the comment and we will be happy to move forward with final decision. 

We look forward to receiving your revised manuscript.

Kind regards,

Shahriar Ahmed, MBBS, MHE, MPhil

Academic Editor

PLOS ONE

Journal Requirements:

**Additional Editor Comments:**

Thank you for submitting the revised version addressing the comments. This manuscript is now almost suitable for acceptance. There is minor comment by reviewer 2 that still needs to be addressed and it should not take more than a few hours of time. Please address the comment and we will be able to mover forward with a decision.

Reviewers' comments:

Reviewer's Responses to Questions

**Comments to the Author**

1. If the authors have adequately addressed your comments raised in a previous round of review and you feel that this manuscript is now acceptable for publication, you may indicate that here to bypass the “Comments to the Author” section, enter your conflict of interest statement in the “Confidential to Editor” section, and submit your "Accept" recommendation.

Reviewer #1: All comments have been addressed

Reviewer #2: All comments have been addressed

2. Is the manuscript technically sound, and do the data support the conclusions?

Reviewer #1: (No Response)

Reviewer #2: Yes

3. Has the statistical analysis been performed appropriately and rigorously? 

Reviewer #1: (No Response)

Reviewer #2: N/A

4. Have the authors made all data underlying the findings in their manuscript fully available?

Reviewer #1: (No Response)

Reviewer #2: Yes

5. Is the manuscript presented in an intelligible fashion and written in standard English?

Reviewer #1: (No Response)

Reviewer #2: Yes

6. Review Comments to the Author

Reviewer #1: The authors have addressed my comments raised in a previous round of review which are accepted by me.

Reviewer #2: Pease add the information on Vietnam's COVID timeline as the epidemic started late (in 2021) compared to other countries as this might have contributed to the difference in addition to what you have mentioned.

7. PLOS authors have the option to publish the peer review history of their article (what does this mean?). If published, this will include your full peer review and any attached files.

Reviewer #1: No

Reviewer #2: **Yes: **Dr Mohammed A Yassin

---

## [Author Response · Author response to Decision Letter 1]

4 Sep 2024

In response to request of reviewer #2, information on Vietnam's COVID timeline added to the resubmitted manuscript on 4 September 2024.

---

## [Editor Report · Decision Letter 2]

6 Sep 2024

Introducing BPaL: experiences from countries supported under the LIFT-TB Project

PONE-D-24-06032R2

Dear Dr. Wares,

We’re pleased to inform you that your manuscript has been judged scientifically suitable for publication and will be formally accepted for publication once it meets all outstanding technical requirements.

Kind regards,

Shahriar Ahmed, MBBS, MHE, MPhil

Academic Editor

PLOS ONE
---

## [Editor Report · Acceptance letter]

13 Sep 2024

PONE-D-24-06032R2 

PLOS ONE

Dear Dr. Wares, 

I'm pleased to inform you that your manuscript has been deemed suitable for publication in PLOS ONE. Congratulations! Your manuscript is now being handed over to our production team.

Kind regards, 

on behalf of

Dr. Shahriar Ahmed 

Academic Editor

PLOS ONE